# Biocatalytic stereocontrolled head-to-tail cyclizations of unbiased terpenes as a tool in chemoenzymatic synthesis

Andreas Schneider [1], Thomas B. Lystbæk[1], Daniel Markthaler[2], Niels Hansen [2] & Bernhard Hauer [1] ✉

Terpene synthesis stands at the forefront of modern synthetic chemistry and represents the state-of-the-art in the chemist's toolbox. Notwithstanding, these endeavors are inherently tied to the current availability of natural cyclic building blocks. Addressing this limitation, the stereocontrolled cyclization of abundant unbiased linear terpenes emerges as a valuable tool, which is still difficult to achieve with chemical catalysts. In this study, we showcase the remarkable capabilities of squalene-hopene cyclases (SHCs) in the chemoenzymatic synthesis of head-to-tail-fused terpenes. By combining engineered SHCs and a practical reaction setup, we generate ten chiral scaffolds with >99% *ee* and *de*, at up to decagram scale. Our mechanistic insights suggest how cyclodextrin encapsulation of terpenes may influence the performance of the membrane-bound enzyme. Moreover, we transform the chiral templates to valuable (mero)-terpenes using interdisciplinary synthetic methods, including a catalytic ring-contraction of enol-ethers facilitated by cooperative iodine/lipase catalysis.

Chemoenzymatic hybrid synthesis is an aspiring paradigm that synergistically merges the advantages of biocatalysis and state-of-the-art synthesis to streamline the synthesis of complex (natural) molecules[1,2]. In their pursuit to generate meroterpenes, the Renata group demonstrated the power of chemoenzymatic synthesis by successfully implementing the regio- and stereoselective remote oxyfunctionalization ability of oxygenases into their synthesis routes[3]. This development opened up entirely new retrosynthetic considerations in terpene synthesis by expanding the available scaffolds from the chiral pool[4]. However, the "chiral pool"[5] or "scaffold remodeling" (SR)[6] strategy is still limited to a few naturally occurring cyclic precursors (see ref. 5) and it often requires multiple chemical steps to transform them into the actually desired natural scaffold which is incorporated into the final product (Fig. 1a and Supplementary Fig. 1a). It is therefore not surprising, that despite their huge potential as bioactive compounds, terpenes are still highly underrepresented e.g., in Pharma (1.3% of APIs)[7].

To overcome this dichotomy, the desired natural terpene scaffold may be forged e.g., by a Brønsted-acid catalyzed stereoselective head-to-tail cyclization of naturally abundant linear precursors within a single step. In this regard, chemists developed various most diverse and creative strategies employing catalysts such as Lewis-Base assisted Brønsted-acids[8], transition metals[9], or supramolecular cages[10]. However, these strategies usually require alternative initiation motifs and/or strong nucleophiles as terminating groups for a successful cyclization (Supplementary Fig. 1b)[11]. Meanwhile the activation of olefins, such as natural unbiased terpenes, via asymmetric Brønsted-acid catalysis has been resolved by List and coworkers[12], however, achieving their stereoselective polycyclization remains challenging (Supplementary Table 1). Nature has employed this strategy for eons[13], utilizing the supramolecular machinery of class II cyclases in the modular biogenesis of terpenes. However, this catalytic excellence was designed to operate on specific substrates and under physiological conditions.

[1]Institute of Biochemistry and Technical Biochemistry, University of Stuttgart, Stuttgart-Vaihingen, Germany. [2]Institute of Thermodynamics and Thermal Process Engineering, University of Stuttgart, Stuttgart-Vaihingen, Germany. ✉e-mail: Bernhard.hauer@itb.uni-stuttgart.de

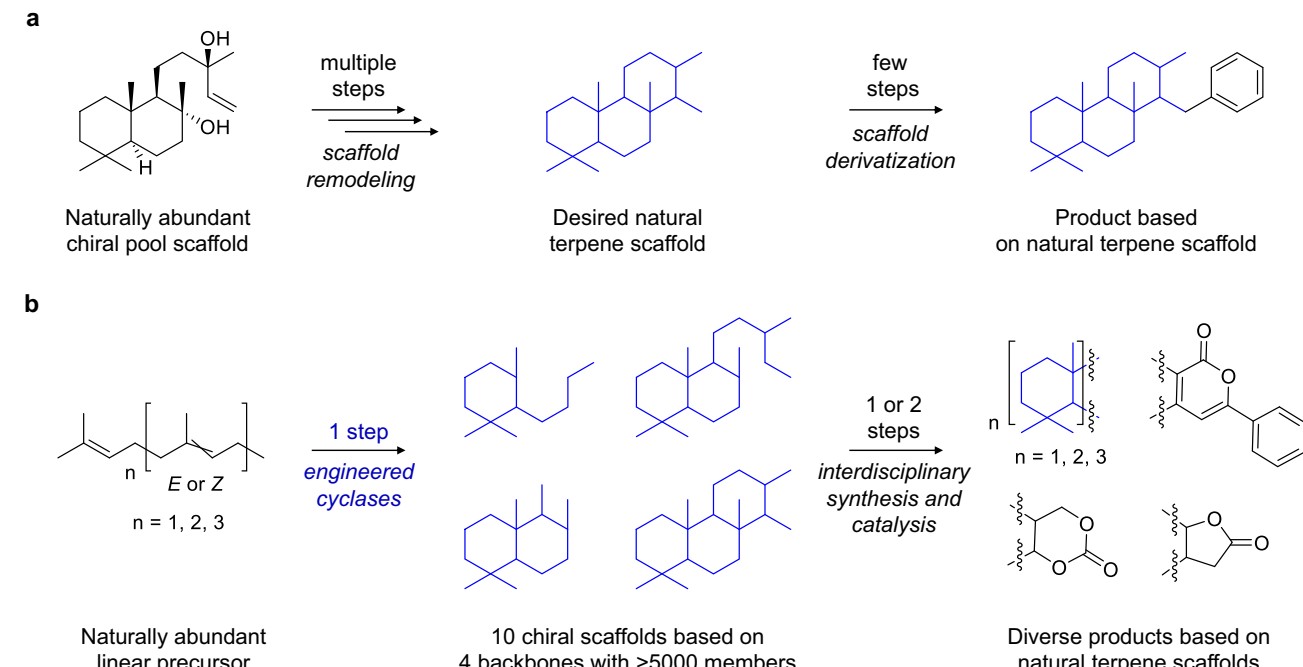

**Fig. 1 | Providing chiral terpene scaffolds via biocatalytic stereocontrolled head-to-tail cyclization. a** The "chiral pool" or 'scaffold remodeling' strategy relies on a few naturally occurring templates, which are successively transformed into the desired natural terpene scaffold via multiple chemical steps[5]. **b** This work: A chemoenzymatic strategy that substitutes the scaffold remodeling by a single biocatalytic stereoselective cyclization to access the desired terpene scaffold. Application of this strategy is demonstrated in the generation of diverse natural scaffolds with >5000 members[40] and consequent transformation towards diverse bioactive products aided by interdisciplinary synthesis and catalysis.

At the outset of this work, the promiscuity and application strategies of SHCs for specific targets have been demonstrated, which usually require tedious isolation protocols or huge host cell amounts (~375 g/L)[14,15]. It should be noted that during the preparation of the manuscript[16] a study by Xiao et al. appeared that also applies the herein presented strategy for terpene synthesis based on the *ent*-isocopolane scaffold[17]. However, a comprehensive study that enables broader integration of this chemoenzymatic strategy has yet to be addressed. By elaborating on a straightforward application and engineering protocol of squalene-hopene cyclases as well as showcasing the stereoselective synthetic potential for diverse bioactive products, we aim to pose this challenge. In this work, we report on enabling the exquisite biocatalysis of the squalene-hopene cyclase (SHC) for streamlined and target-oriented chemoenzymatic synthesis of terpenes (Fig. 1b).

## Results

### Generation of cyclic terpene precursors by engineered SHCs

To begin, we selected terpene targets from diverse areas of application and identified their inherent cyclic moiety (Fig. 2a, blue structures). Actinidiolide derivatives **1** are prominent bioflavors and insect pheromones, and are built from a C1 elongated monocyclic cyclogeraniol skeleton[18]. Biopolymers based on 1,3-dioxanone **2** include the drimane diol skeleton and are used e.g., in tissue engineering[19]. Meroterpenes as metachromins **3** are applied in e.g., cancer treatment[20] studies and incorporate the C13 megastigmane skeleton. Another representative of this compound family is the sponge-derived meroterpene **4** that bears potential in inflammation disease treatment[21] and is made of the tricyclic *ent*-isocopolane skeleton. The last and most complex targets, the α-pyrone meroterpenes **5**, offer a promising broad range of biological activities[22] and can be derived from a sclareoloxide-like β-keto ester structure. We are aware that the targets presented herein lack functionalizations, such as the OH-group at position C-3 of **5**, which emphasizes the important contributions of the Baran[23] and the Renata group in the field of terpene oxidations[3]. However, the focus of this work was to showcase the synthetic potential of biocatalytic

cyclizations *en* route to terpenes, which is why prior, or late-stage oxidations were not considered.

With these retrosynthetic analyses in mind, we commenced screening our in-house SHC library with the appropriate linear terpenes, starting with the smallest one E-geranyl acetate **6** (Fig. 2b). SHC variants were cultivated in 96DW plates and biotransformations were carried out in vivo as this setup proved its efficiency for the membrane-bound enzyme[24]. Fig. 2c shows the AacSHC wildtype compared with the two most interesting hits for each substrate. Moreover, to learn from the substrate-mutant relation, all substrates were evaluated in silico by docking them into the confined active site of the AacSHC wildtype (Fig. 2d and Supplementary Fig. 2). The stereoselective cyclohydration of **6** would directly provide the precursor cyclogeranyl acetate hydrate **7** for the synthesis of **1** and thus overcome a three-step SR[18]. To our delight, variant G600R generated cyclohydration product with 95% selectivity and boosted the WT activity sevenfold.

Cyclohydration of E,E-farnesol **8** would result in the desired drimane skeleton for dioxanone **3** formation and shorten concurrent SR routes by two steps[25]. While this substrate has already been the target of AacSHC-mediated cyclization by Hoshino and Hauer[26,27], those studies employed purified enzymes and identified variants lacked full selectivity and conversion of **8**, respectively. Among multiple hit positions (cf. Supplementary Fig. 4), variant A306W achieved high conversion (89%) and excellent drimane diol **9** selectivitiy (94%). Notably, we found variant L36V/Y420F/G600L that enabled the stereocontrol of the final hydration and provided the opposite diastereoisomer **S1**–**9** with a diastereoselectivity of 88% (Supplementary Fig. 4 and NMR data).

In view of the sponge-derived meroterpene **4**, we next focused on the cyclization of linear diterpene E,E,E-geranyl geraniol **10**. Hoshino and coworkers reported the promiscuous tricyclization of **10** to **11** using purified AacSHC WT, albeit with a low yield of 12%[27]. and biocatalytic access to labdane diterpenes was reported but required the extraction of 3 L fermentation broth and yielded only 3 mg product[28].

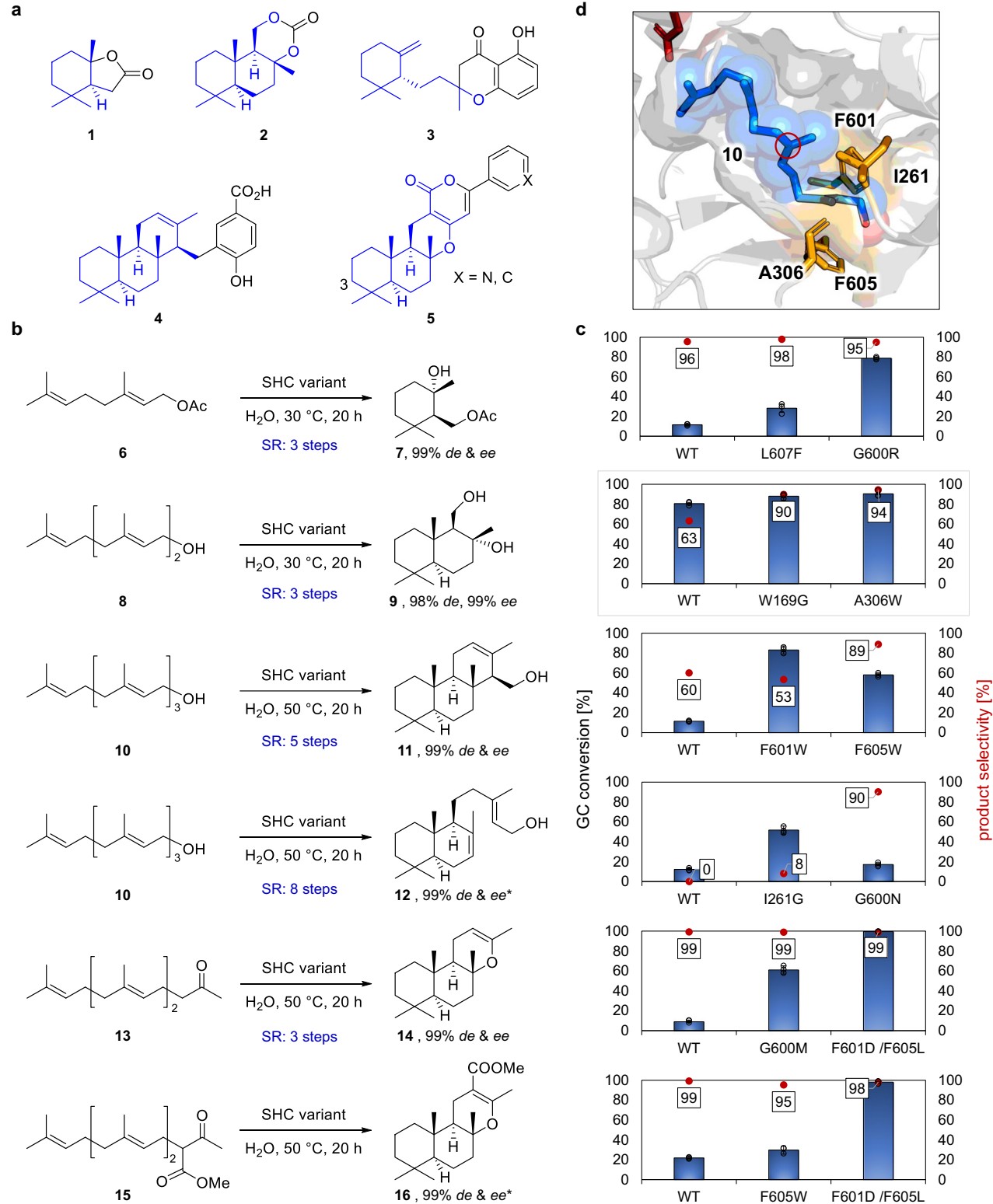

**Fig. 2 | Terpene targets and accessing their terpene skeleton by engineered SHCs. a** Selected cyclic terpenes 1–5 from different areas of application[18–22,53,54], their terpene skeleton (blue), and the additionally required functionality (black). **b** Linear terpene precursors selectively cyclized by engineered SHC variants to the desired cyclic scaffolds in analytical biotransformations. SR scaffold remodeling. * assumed due to shape-complementary substrate prefolding in the SHC active site (Supplementary Fig. 2) and circular dichroism data. Please see supporting information for reaction details. **c** GC conversion and product selectivities of the best hits compared to the AacSHC wildtype (WT) as

determined from the area of the GC-FID peaks (cf. Supplementary Figs. 3–9). **d** Substrate 10, shown as blue sticks and spheres, docked in the confined active site of the AacSHC (PDB: 1UMP), exemplifies the generally observed prefolding of all substrates. Most identified beneficial mutations are positioned in a sphere around the terpene's functional group (orange sticks). This can be leveraged as a mutagenesis strategy of these enzymes. Reactions were performed in technical replicates $n = 3$ (dot plots). Bars represent mean values ± SD (see SI for more details). Source data are provided as a Source Data file.

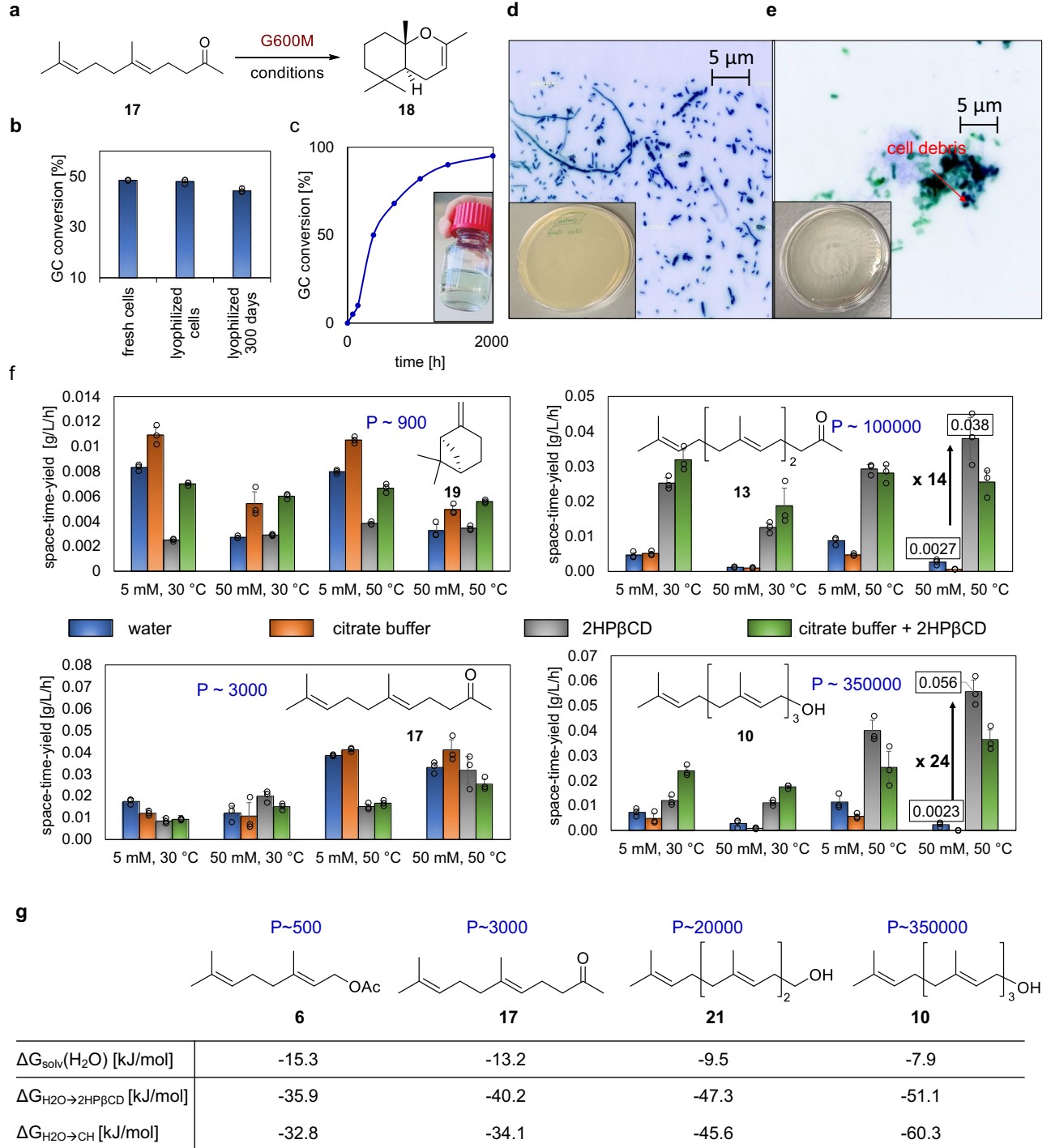

**Fig. 3 | Establishing a concise and comprehensible setup for scaling up "in vivo" SHC biocatalysis. a** Model reaction for initial experiments. **b** Comparison of freshly expressed cells, lyophilized cells, and stored lyophilized cells by employing the model reaction. **c** Time-conversion curve of the model reaction in a 5 L reactor and the isolated product. **d** Freshly expressed E. coli cells after 16 h of incubation on an Agar-plate and under the fluorescence microscope show highly abundant and viable cells (please see Supplementary Fig. 11 for more details). **e** Lyophilized E. coli cells after 16 h incubation on an Agar-plate and under the fluorescence microscope show no growth on the plate and predominatly cell membrane debris. **f** Space-time-yield of analytical biotransformations using (+)-β-pinene 19 (TelSHC C312S), E-geranyl acetone 17 (AacSHC G600M), E,E-farnesyl acetone 13 (AacSHC F601D/F605L), and E,E,E-geranyl geraniol 10 (AacSHC F605W) under varying substrate

concentrations and temperatures in water (blue bar), 100 mM citrate buffer (pH = 6.0, orange bar), 50 mM 2-hydroxypropyl-β-cyclodextrin (2HPβCD, gray bar), and 100 mM citrate buffer (pH = 6.0) + 50 mM 2HPβCD (green bar). Partition P = [octanol/water] calculated by ChemDraw is given as blue numbers (see Supplementary Fig. 13 for exact numbers). High number means high hydrophobicity. Reactions were performed in technical replicates $n = 3$ (dot plots). Bars represent mean values ± SD (see SI for more details). **g** Free energy calculations of the transfer free energy of substrates 6, 17, 21, and 10 from water into 2HPβCD and cyclohexane set as an artificial membrane (for justification, see computational methods in the supporting information). For reaction details of analytical biotransformations, calculation parameters, and controls, please see SI. Source data are provided as a Source Data file.

Our aim, therefore, was to direct the cyclization of **10** towards both products with high selectivity and high conversion using engineered SHCs. Regarding the tricyclization, beneficial mutations were identified at positions F601 and F605. It turned out that most of the hits, e.g., F601W were more active (up to 8-fold) than the WT but lacked selectivity towards **11** (Supplementary Fig. 5). However, variant F605W produced the α-product ent-isocopolol **11** with fivefold improved conversion and ~90% selectivity. Favoring bicyclization of **10** was achieved by mutations at I261, G600, and L607 that are located around the third transient carbocation (Fig. 2d, red circle). Congruent to the tricyclization, most variants yielded product mixtures which emphasizes the challenge of selective cyclizations even with an enzyme (Supplementary Fig. 5). Merely, the asparagine at position 600 resulted in high (90%) selectivity towards the bicyclic labdane scaffold **12** with a conversion comparable to the WT. Interestingly, the in silico docked product **12** in computationally generated variant G600N gives rise to a dual function of the asparagine that is anchoring the functional group[29] and acting as a Brønsted-base (Supplementary Fig. 6)[30]. Laboratory efforts towards ent-isocopolol **11** and labdanol **12** encompass 5 and 8 steps, respectively[31,32].

Coming to our last target, the α-pyrone meroterpenes **5**, we drew inspiration from a study by ref. 33, which used a non-natural sclareoloxide-like terpene structure that was racemically cyclized using electrophilic mercury. In preparation for this transformation, we evaluated the cyclization of E,E-farnesyl acetone **13** to sclareoloxide **14** with our SHC library. We chose this strategy due to the fact that the natural substrate **13** and the non-natural linear precursor **15** are almost identically pre-folded in the active site of the AacSHC WT (cf. Supplementary Fig. 3e, f). Hit variants for **13** should then be tested with **15** to save time and resources. Promiscuous cyclization of **13** towards sclareoloxide **14** with purified AacSHC has been reported, albeit with very low conversions below 1%[34]. Our survey yielded variant F601D/F605L, among multiple other hits (Supplementary Fig. 7), which surpassed the WT tenfold, while ensuring high selectivity. Curiously, the amino acid exchanges F601D and F605L both presumably result in less confinement around the keto-group of **13** in the active site, which is contrary to the increased conversion at first glance. However, the introduced aspartate may anchor the keto-group and, therefore, lock the substrate in the right prefolding faster or act as Brønsted-base to activate the final nucleophile. Subsequently, we used the non-natural linear precursor **15** with the distinguished SHC variants. Notably, we noticed that **15** is prone to decarboxylate to **13** (Supplementary Fig. 8), thus the Brønsted-acid catalyst has to overcome this side reaction and, moreover, select one out of three potential final nucleophiles. Intriguingly, variant F601D/F605L exhibited excellent conversion (99%, fivefold higher compared to WT) and selectivity (98%) towards the desired cyclic product **16**, which emphasizes the reliability of our docking results and highlights the precise catalyst control of the SHC. Cyclic **14** can be prepared by SR in three steps[35], whereas **16** has only been racemically cyclized yet.

In summary, we were able to produce all desired (and more in the SI) terpene scaffolds with high conversions and excellent stereoselectivities, empowered by the tunable shape-complementary prefolding of substrates in the confined active site of the biocatalyst. It should be noted that the identified hits originate from earlier published studies[24,29,36–38]. Our data demonstrate that mechanism-guided enzyme engineering in the sphere around the desired transient carbocation, as showcased for substrate **10** in Fig. 2d, enables the direction of cationic head-to-tail cyclizations in terms of regioselective α-/γ-deprotonation, stereoselective hydration, and cascade progress. Leveraging this knowledge, which builds on our prior findings[29,37,39], paves the way to the cyclization of dozens of carbon skeletons with divergence potential to tens of thousands of natural products[40]. To the best of our knowledge, there is no chemical catalyst that is able to control a stereoselective one-step cyclization of these unbiased

terpenes to such an extent (cf. Supplementary Table 1 and supplementary chromatograms), especially with an alkene as the terminal nucleophile (substrates **6**, **8**, **10**). A limitation of the presented strategy is that, surprisingly, no single β-deprotonation product could be detected.

## Technical and mechanistic investigations on the biocatalyst preparation

Having ascertained the stereocontrolled cyclization, we intended to provide a concise setup to translate SHC biocatalysis from lab to liter scale. As a model reaction for initial investigations, we chose the broadly studied promiscuous cyclization of E-geranyl acetone **17** using AacSHC variant G600M[29] (Fig. 3a). First, we proved the ability to use lyophilized E. coli whole-cells, which would drastically simplify the application of SHCs as a storable powder. To our delight, biotransformations showed no difference in conversion, and long-term stability (Fig. 3b). To improve substrate availability, four cyclodextrins were tested, which disclosed 2-hydroxypropyl-β-cyclodextrin (2HPβCD) as the best candidate in terms of substrate conversion (Supplementary Fig. 9). Next, we proved scalability with a previously[29] elaborated protocol using 5 g/L (25 mM) substrate, 10 g$_{CDW}$/L cells, 14 g/L (10 mM) 2HPβCD, and buffer in a 5 L reactor stirring with 100 rpm at 30 °C what demonstrated a slow but operationally stable catalyst for 84 days and yielded 22.4 g (90%) of cyclic **18** (Fig. 3c). We ascribed the operational stability to the fact that SHC does not require a viable cell, but parts of the cell membrane, where the enzyme is embedded (see Thermolysis protocol in the supporting information), are enough to drive the biocatalysis. This hypothesis was strengthened by comparing the growth of freshly expressed with the lyophilized cells on an Agar-plate, which disclosed that few cells survive the lyophilization process. Further evidence was provided by fluorescence microscopy which showed that mainly cell debris remains (cf. Fig. 3d, e and Supplementary Fig. 10).

Intrigued by this data, we next examined parameters that may influence the space-time yield of the biocatalysis. Therefore, buffer, cyclodextrin, and the combination thereof were evaluated as additives. Moreover, terpene type (as defined by their partition coefficient P = [octanol/water], Fig. 3f, blue numbers), terpene concentration, cell concentration, and temperature were evaluated to get a coherent picture (Fig. 3f). SHC variants TelSHC C312S[41], AacSHC G600M[29], AacSHC F601D/F605L and AacSHC F605W, were used with substrates **19**, **17**, **13**, and **10**, respectively, for the biocatalysis.

While the more hydrophilic terpenes (+)-β-pinene **19** and E-geranyl acetone **17** were better converted to (+)-α-pinene **20** and trans-hexahydrochromene **18** in the absence of 2HPβCD, the more hydrophobic substrates **13** and **10** were generally better cyclized in its presence. Vice versa the chelating citrate buffer improved the cyclization of the more hydrophilic substrates **19** and **17** (cf. Supplementary Fig. 11 for non-chelating buffer), however, impeded the cyclization of more hydrophobic ones. In the worst case, using buffer completely disrupted the reaction (see **10** at 50 °C, 50 mM). The combination of both or increasing the cell density did not show clear trends in either direction (Supplementary Fig. 12). Intuitively, the beneficial effect of 2HPβCD rises with increasing P of the substrates (Supplementary Fig. 13) and, finally, it is generally advisable to employ a thermostable enzyme variant at higher substrate concentrations, which is known correlation[42] (Supplementary Fig. 14).

Tempted by the substrate-dependent effects of 2HPβCD in our setup, we calculated the transfer free energies of the substrates geranyl acetate **6**, geranyl acetone **17**, homofarnesol **21**, geranyl geraniol **10** from water into the core of 2HPβCD (ΔG$_{H_2O→2HPβCD}$) versus the membrane (ΔG$_{H_2O→CH}$), where the enzyme naturally sources its substrates, using the double decoupling method (for more information and selected calculated products, such as ambroxide **22** see Supplementary Table 7). The membrane was not

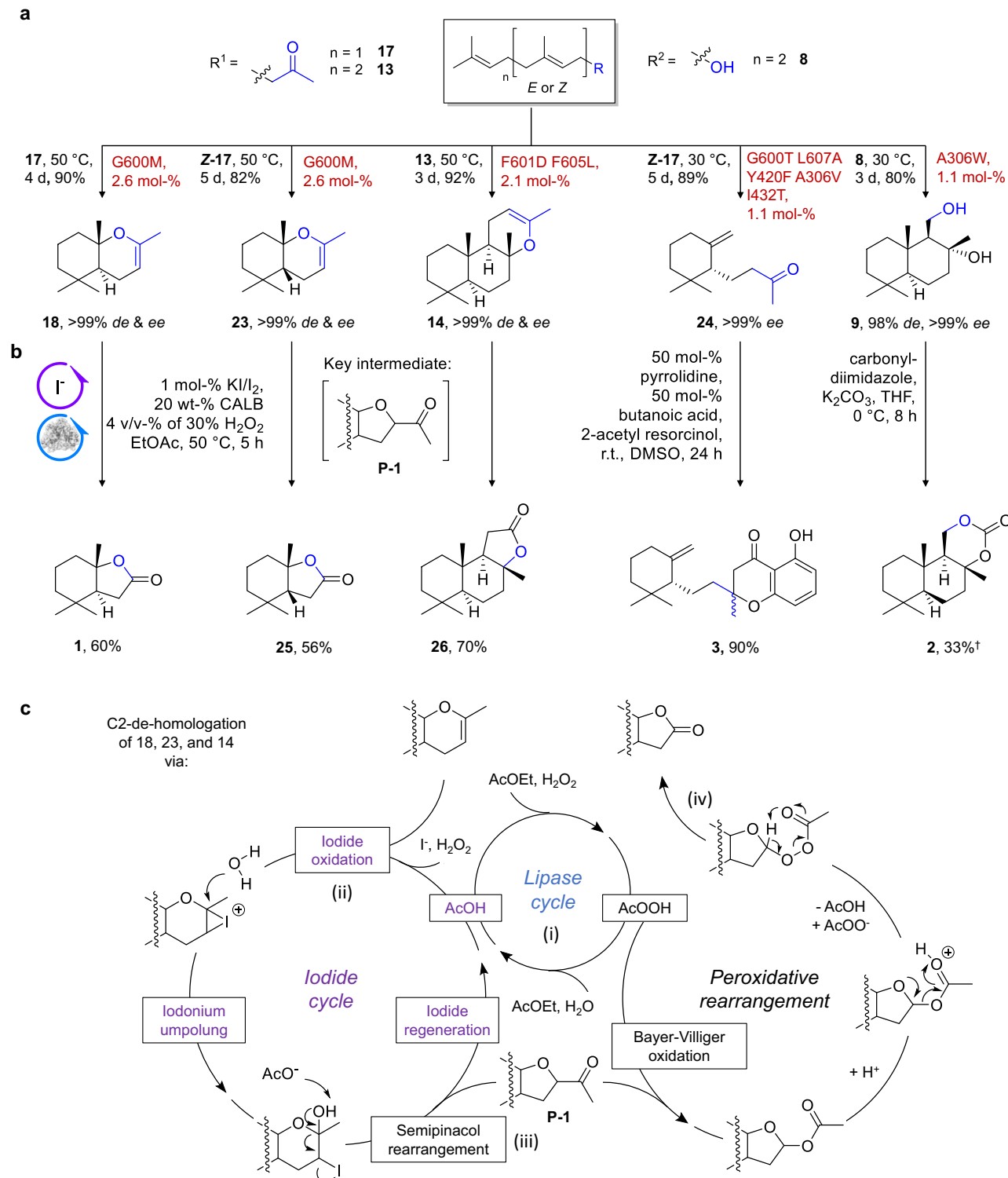

**Fig. 4 | Hybrid syntheses of terpenes employing SHC biocatalysis (part 1).**
**a** Module 1, scaffold diversification: Biocatalytic generation of chiral cyclic terpenes 18, 23, 14, 24, and 9. **b** Module 2, scaffold derivatization: Interdisciplinary synthesis and catalysis to derivatize the chiral templates. **c** One-pot chemoenzymatic C2-dehomologation strategy of cyclic enolethers 18, 23, and 14 employing iodine and lipase catalysis (cf. Supplementary Figs. 16, 17). For upscaling conditions and synthesis details please see supporting information. †NMR yield.

modeled explicitly but instead mimicked by bulk cyclohexane based on the good agreement between calculated free energies for transferring amino acid side chains from water into the center of a lipid bilayer and the experimental water to cyclohexane transfer free energy[43]. This data disclosed that (a) membrane partition, and

encapsulation of terpenes by 2HPβCD from the water phase is generally beneficial for the system (cf. $\Delta G_{solv}(H_2O)$ and transfer free energies), (b) both processes are competing, and (c) 2HPβCD encapsulation of more hydrophilic substrates **6**, **17**, **21** is slightly stronger than their transfer into the membrane or almost equal. It

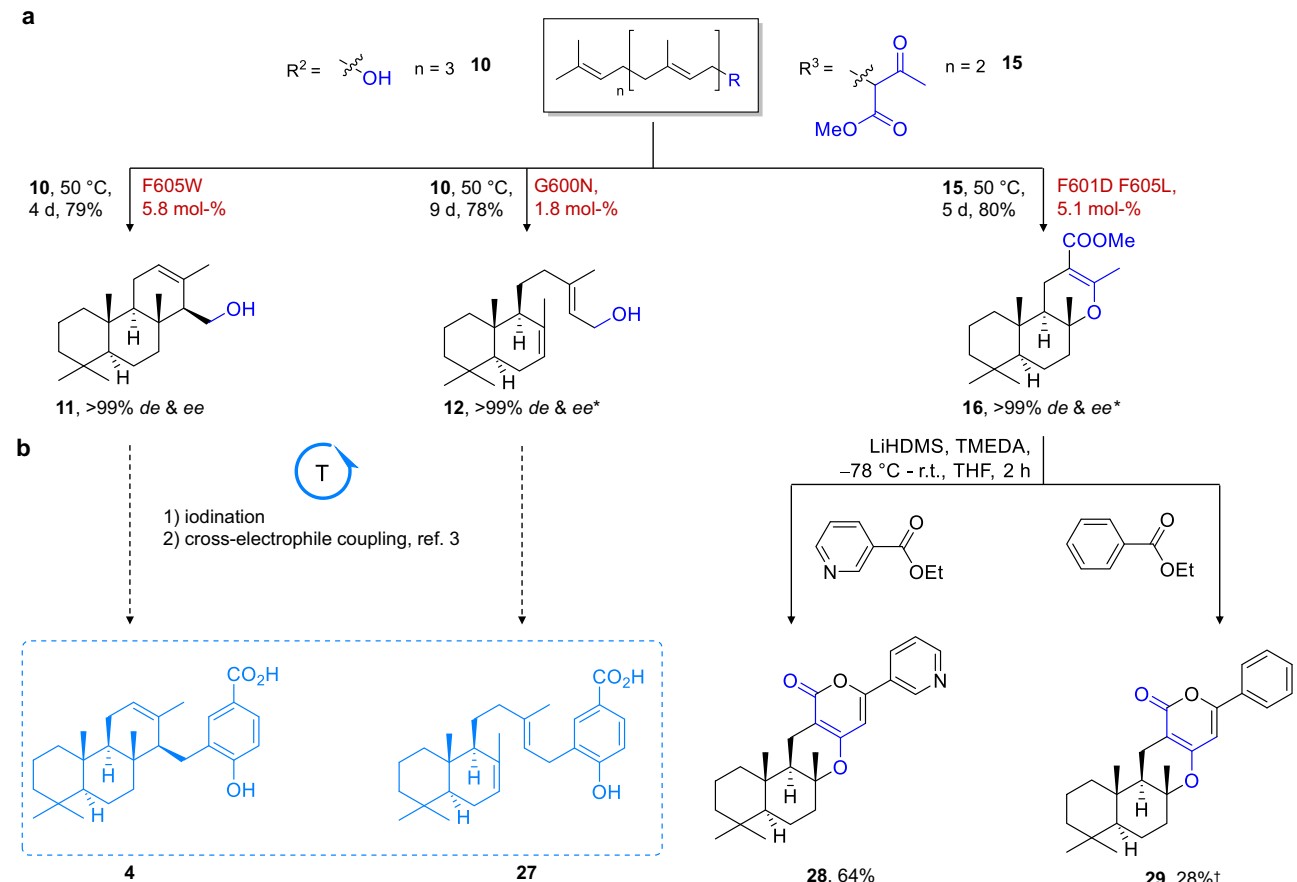

**Fig. 5 | Hybrid syntheses of terpenes employing SHC biocatalysis (part 2).**
**a** Module 1, scaffold diversification: Biocatalytic generation of chiral cyclic terpenes 11, 12, and 16. **b** Module 2, scaffold derivatization: Potential strategy for meroterpenes 27 and 4, as well as the herein applied tandem γ-acylation/ intramolecular annulation to access α-pyrone meroterpenes 28 and 29. * assumed due to shape-complementary substrate prefolding in the SHC active site (Supplementary Fig. 2) and circular dichroism data. Please see supporting information for reaction details. †NMR yield determined via mesitylene standard.

should be noted that both transfers are reversible processes, as otherwise biocatalysis would be inhibited.

To sum up, our data set disclosed that the SHC is independent of a viable cell, which ensures long-term storage as well as operational stability. However, the setup is largely dependent on the sum of abiotic stressors, such as temperature, buffer, and terpenes, acting on the biological system of membrane and membrane-bound enzyme (Supplementary Fig. 15). The pivot herein is to recognize the substrate's level of hydrophobicity and based on that augment biocatalysis by setting the other parameters with special focus on temperature and internal organic reservoirs such as cyclodextrin. Presumably, the encapsulating host not only improves the solubility of the more hydrophobic molecules but also weakens their hydrophobic effect on the biological system by reversible encapsulation. These findings emphasize the practicability and flexibility of the biocatalyst setup for synthetic purposes.

The elaborate substrate-focused protocol thus paved the way for the hybrid syntheses of (mero-)terpenes devised in Fig. 2a. Tetrahydroactinidiolides, such as **1**, are usually prepared via C1 homologation of the appropriate terpene such as **7**[18]. Conversely, we envisioned a catalytic strategy that readily de-homologizes cyclic enolethers, such as **18**, (Fig. 4a, b) inspired by the work of ref. 35. First, we examined this transformation in two partial reactions: an iodine-mediated and a peroxidative rearrangement using enolether **18** (Supplementary Fig. 16). Evaluation of the iodine reaction conditions disclosed that catalytic (5 mol-%) amounts of sodium iodide in the presence of excess $H_2O_2$ (successively added) and acidic buffer are

enough to yield semi-pinacol rearrangement product **P-1** (60:40 dr) (Fig. 4c) almost quantitatively (97%). For the peroxidative rearrangement, we employed candida antarctica lipase (CALB) that can generate peracetic acid from $H_2O_2$ and ethyl acetate (EtOAc)[44], which served as the solvent simultaneously. As stated by the authors of ref. 35, the reaction had to be carried out at 50 °C and yielded lactone **1** with 70% in the best scenario (Supplementary Fig. 16). Conveniently, the oxidative rearrangement consumes both semi-pinacol diastereoisomers (Supplementary Fig. 17). Finally, we combined both reactions in one pot using sclareoloxide **14** as the substrate EtOAc as the solvent and substituting sodium iodide with lugol's iodine (KI/I₂) to overcome solubility issues of the iodide catalyst. Careful evaluation of the reaction conditions led to the optimal setup consisting of 50 mM substrate shaken in EtOAc, 1 mol% Lugol's iodine, 20 wt% CALB, and 4 v/v% of 30% $H_2O_2$ (10 eq., successively added) at 50 °C for 5 h. Our proposed mechanism is depicted in Fig. 4c: Initially (i), CALB generates acetic and peracetic acid (AcOH and AcOOH) from water and EtOAc, which in combination with $H_2O_2$ is used for the acidic-oxidative generation of "I⁺" (ii) (not further specified as iodine species are in a complex equilibrium)[45]. The subsequently formed iodonium intermediate is then nucleophilically attacked by water to form a semiacetal, which immediately generates ketone **P-1** after deprotonation and semi-pinacol rearrangement (iii). **P-1** then forms a Bayer-Villiger product in the presence of AcOOH, generated by CALB, EtOAc, and $H_2O_2$. After protonation, rearrangement, and cleavage of AcOH, the transient oxonium-ion reacts with AcOO⁻, which finally rearranges to form lactone **1** (iv).

Employing SHC biocatalysis, we generated **17**, **Z-17**, and **13** to chiral enolethers **18**, **23**, **14** with high yields (82-92%) and subsequently edited their skeletal constitution using the iodine/lipase protocol to form trans-tetrahydroactinidiolide **1**, cis-tetrahydroactinidiolide **25** and (+)-sclareolide **26** with moderate yields of 56-70% in one pot. Next, (−)-γ-dihydroionone **24**, which was prepared via directed cationic cyclization of **Z-17** with 89% yield, was coupled with 2-acetyl resorcinol in a pyrrolidine-catalyzed tandem aldol/ intramolecular Michael addition, i.e., the Kabbe reaction[46], to generate chromanone **3** with 90% yield and 50:50 dr. Solid drimen diol **9** was cyclized with a yield of 88% and transformed into carbonate **2** via potassium carbonate ($K_2CO_3$) mediated carbonylation with a moderate yield of 33%. Labdane **12** and ent-isocopolane **11** were also generated with high yields of 78% and 90%, respectively, from linear **10** and could potentially be transformed within three steps to sponge-derived meroterpenes **4** and **27** e.g., by transition metal-catalyzed cross-electrophile coupling of iodinated **12** and **11** with iodinated methoxy-benzoic esters as demonstrated in ref. 3. (Fig. 5). Finally, α-pyrone meroterpenes **28** and **29** that constitute the carbon skeleton of pyripyropenes and phenylpyropenes, were generated from chiral **16** and nicotinoic as well as benzoic ester in a base-mediated tandem γ-acylation/ intramolecular annulation reaction with yields of 64 and 28%, respectively (for a summary see Supplementary Table 2).

Conclusively, we could prove that the exquisite catalysis of cyclases can be harnessed for target-oriented synthesis of terpenes. Scaffold remodeling approaches to cyclic terpenes, while being testimonials of chemical creativity, can be shortened by up to 90% to essentially one diastereo- and enantiopure cyclization (cf. Fig. 2b), which is easily applicable as well as scalable, and provides yields up 92% at the same time. Combined with strategic interdisciplinary synthesis and catalysis, we thus provided access to high molecular complexity in only two or three steps. Thus, the stage is set for novel or drastically shortened retrosynthetic logic in de novo terpene synthesis.

## Discussion

The golden age of biocatalysis enabled chemists to harness the catalytic power of enzymes, tailor them, and unlock their synthetic capabilities to streamline access to complex molecules[47,48]. We herein provide the biocatalytic stereocontrolled head-to-tail cyclization as a tool in chemoenzymatic terpene synthesis. As a result, the pool of available cyclic starting building blocks can be broadly expanded and trigger new retrosynthetic considerations. The presented setup using bench-stable cell powder and encapsulating agents simplifies the application of these enzymes to the level of batch chemistry. The key herein is to mimic the new-to-nature conditions on a small scale, and adapt the system to inherent limitations, such as substrate hydrophobicity. Thus, this study also represents a helpful entry in the industrially oriented enzymology recently introduced by Woodley and coworkers[49]. Limitations of the presented strategy are long reaction times for the cyclization (up to 3–20 d), which is probably related to the membrane diffusion issues and has been the target of earlier studies[41]. Moreover, the enantioselectivity is determined by the enzyme's active site, which could be overcome by employing class II cyclases with inverted selectivity[50].

The ideal synthesis encompasses synthetic routes "[…] involving no intermediary refunctionalizations, and leading directly to the target […]"[51]. En route to cyclic terpenes, the stereocontrolled cationic cyclization genuinely epitomizes this overarching synthetic goal. Merging this tool with state-of-the-art synthesis and catalysis, such as electrochemical generation of the linear precursors[52] and biocatalytic remote oxidations[3] will further ease the access to desired head-to-tail-fused terpenes. Moreover, harnessing the synthetic power of class I cyclases, will broadly expand the pool of accessible cyclic terpene scaffolds. Finally, we assume that hybrid strategies that leverage the best of both worlds−natural and man-made tools−will be a strong driving force in the pursuit of the ideal synthesis.

## Methods

### Lyophilization protocol

Freshly harvested *E. coli* cells were transferred to petri dishes and frozen at −80 °C overnight. On the following day the frozen pellets were quickly transferred to Christ alpha 2−4 LD plus lyophilizer and lyophilized at −80 °C at 0.0001 atm for 1 day. The resulting lyophilized whole cells were mortared gently and stored in 50 mL Falcon tubes at room temperature. The enzyme content for each batch was determined using the thermolysis purification protocol.

### Thermolysis purification[25,26]

Lyophilized cells (10 mg) were resuspended in 1 mL Lysis buffer and incubated for 30 min at 70 °C. The cell suspension was centrifuged ($14,000 \times g$, 1 min) and the supernatant was discarded. As the enzyme is membrane-bound, 1 mL CHAPS buffer was added to extract it from the cell pellet by shaking at room temperature for 1–2 d, 600 rpm. After subsequent centrifugation ($14,000 \times g$, 1 min) the supernatant containing the SHC (Aac or Tel) was transferred to a new tube followed by SDS-PAGE analysis and determination of enzyme concentration by using the Nanodrop 1000 (Agilent, Santa Clara, US). Therefore the "Protein A280" mode was chosen with MW = 71439 Da and molar extinction coefficient ε = 185180 as protein-specific data for AacSHC and MW = 72668 and ε = 189230 for TelSHC.

### Analytical biotransformations in GC screw-cap-vials

About 10 mg/ml lyophilized cells were resuspended in water, water supplemented with cyclodextrins or buffer or both. About 495 μL of the cell suspension were transferred to GC screw-cap-vials and 5 μL of a substrate/DMSO stock ($c_{end}$, substrate = 1 mM, unless otherwise noted in the reaction conditions) were added to start the reactions. Reactions were shaken at 30 or 50 °C for 20 h, depending on the substrate (see Fig. 2). Reactions were stopped by adding 600 μL cyclohexane:ethyl acetate (1:1), vortexing, and shaking for 5 min. After centrifugation ($3600 \times g$, 5 min), the organic phase was analyzed via GC-MS equipped with a PAL-Sampler directly from the two-phase system. Quantification was done by GC-FID and dodecane as an internal standard.

### General procedure of preparative scale biotransformation

After the product-selective AacSHC variant was identified, it was produced in *E. coli* on a large scale (6–12 Erlenmeyer flasks) following the protocol described in the SI. The resulting cell pellets were lyophilized afterward, and the enzyme concentration was determined via the thermolysis protocol. The reaction mixture contained a given amount of cells, the given amount of substrate, and optionally a given amount of 2-HPCD or citrate buffer. The reactions were performed in closed 100 mL or 1 L Schott-flasks in an INFORS-HT incubator at 180 rpm for the given time. After the reaction was finished (GC monitoring) the cell suspension was overlayed with 50 or 500 mL cyclohexane:ethyl acetate (1:1) and stirred slowly to avoid mixing of the phases for 24 h. The organic phase was dried over $MgSO_4$, reduced, and the crude products were analyzed via GC prior to column chromatography (20:1 cyclohexane:ethyl acetate). GC yield was determined by [area$_{product}$]:[area$_{substrate}$+area$_{product}$].

### Fluorescence microscopy

Samples for Fluorescence microscopy were prepared as follows: 1 mg$_{CDW}$ lyophilized and 4.5 mg$_{CWW}$ of freshly prepared *E. coli* cells were fixed using 100 μl 4% paraformaldehyde in PBS solution. In order to wash the cells, the mixture was centrifuged ($12,000 \times g$, 30 s), the supernatant was discarded, and 1 mL of PBS solution was added. The cells were resuspended and centrifuged ($12,000 \times g$, 30 s) again. The supernatant was discarded, and the resulting pellet was resuspended

on 100 µl sterile $H_2O$. Afterward, 5 µl of the washed samples were struck out on a glass plate (Thermo Scientific) marked with 5 µl DAPI (Sigma-Aldrich) and SYTOX™ 4-64 (Thermo Scientific) and visualized using a Fluorescence microscope (ZEISS Axiovert 200 equipped with a Axio-Cam HRm) with filters: Fs01 365 ± 12 nm and Fs09 470 ± 40 nm. Pictures were edited with ImageJ. The resulting pictures are presented in Fig. 3 and Supplementary Fig. 10.

### Reporting summary
Further information on research design is available in the Nature Portfolio Reporting Summary linked to this article.

## Data availability
All data were available in the article and its Supplementary Information file; data are also available from the corresponding authors upon request. The squalene-hopene cyclase protein data used in this study are available in the UniProt database under accession code P33247. Source Data are provided as a Source Data file. Input files required to run the molecular dynamics simulations are available at https://doi.org/10.18419/darus-4135. Source data are provided with this paper.

## Code availability
Molecular dynamics simulations in this work were performed using the open-source code GROMACS patched to the open-source library PLUMED.

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

## Acknowledgements

The authors thank Sven Richter for providing the reactor and Wendy Escobedo for introducing it into Fluorescence microscopy. The authors also thank all early readers for their helpful comments. A.S. and B.H. gratefully acknowledge the Deutsche Forschungsgemeinschaft (DFG HA 1251/6-1) for research funding. NH thanks the Deutsche Forschungsgemeinschaft (DFG, German Research Foundation) for supporting this work by funding—EXC2075—390740016 under Germany's Excellence Strategy. N.H. and D.M. acknowledge the support by the Stuttgart Center for Simulation Science (SimTech), the High Performance and Cloud Computing Group at the Zentrum für Datenverarbeitung of the University of Tübingen, the state of Baden-Württemberg through bwHPC and the DFG through grant no INST 37/935-1 FUGG.

## Author contributions

A.S. designed the project. A.S. and B.H. supervised the project. A.S. and T.B.L. performed enzymatic reactions, fluorescence microscopy and upscaling studies. AS conducted substrate and product synthesis. D.M. and N.H. conducted free energy calculations.

## Funding

## Competing interests

The authors declare no competing interests.
