## [Peer Review File · Nature Communications]

Biocatalytic stereocontrolled head-to-tail cyclizations of unbiased terpenes as a tool in chemoenzymatic synthesisEditorial Note: This manuscript has been previously reviewed at another journal that is not operating a transparent peer review scheme. This document only contains reviewer comments and rebuttal letters for versions considered at *Nature Communications*.

REVIEWER COMMENTS

Reviewer #1 (Remarks to the Author):

This revised manuscript meets all my previously expressed concerns.

Reviewer #2 (Remarks to the Author):

This revised manuscript by Hauer and co-workers is a significant improvement from the initial submission and has addressed several key points raised by the previous round of review. Overall, this referee is now generally in favor of acceptance of this manuscript in *Nature: Communications*, though was slightly disappointed to find in some cases that the thoughtful points raised from that previous round of review were ignored, especially in terms of extending substrate scope given the publication of a related paper in the literature. At this point, however, this referee believes that the final area where additional improvement is still needed is with the spectra. As noted by two reviewers, several impurities existed within the NMR spectra of the original submission. That situation remains now, noting compounds 24, 14, S1-9, and 11 as just a few of many that have clear additional materials as a result of several non-enumerated peaks. Critical to this referee is that the authors comment on their yield calculations based on these impurities and/or delineate their presence directly in some way. Some points of confusion in the experimental remain, such as on page 44 the formation of the acetate makes little sense, especially as 1:1 impurity. The authors could merely have quenched their reaction differently, and or monitored by TLC, to ensure that an intramolecular cyclization proceeded smoothly, noting again that such products can arise directly via carbamate groups attached to the terminal alcohol of the starting materials.

Finally, it is also recommended the authors consider the precedents from Yamamoto and

Corey for chiral Bronsted acid-promoted cyclizations as part of their opening analysis, not just the List paper, as many examples of this type exist and has been a problem that others were true pioneers in terms of initial examples. Better review citations might help here in the beginning, as again the authors ignore a lot of key scholarship.

a) Ishihara, K.; Nakamura, S.; Yamamoto, H. J. Am. Chem. Soc. 1999, 121, 4906. (b) Nakamura, S.; Ishihara, K.; Yamamoto, H. J. Am. Chem. Soc. 2000, 122, 8131. (c) Ishihara, K.; Ishibashi, H.; Yamamoto, H. J. Am. Chem. Soc. 2001, 123, 1505. (d) Ishihara, K.; Ishibashi, H.; Yamamoto, H. J. Am. Chem. Soc. 2002, 124, 3647. (e) Ishibashi, H.; Ishihara, K.; Yamamoto, H. J. Am. Chem. Soc. 2004, 126, 11122. (f) Surendra, K.; Corey, E. J. J. Am. Chem. Soc. 2012, 134, 11992. (g) Surendra, K.; Rajendar, G.; Corey, E. J. J. Am. Chem. Soc. 2014, 136, 642.

Overall, again, this submission is an improvement, and key results are obtained in this work; clarity of outcome is the main desire remaining as noted based on the points raised.

Reviewer #3 (Remarks to the Author):

In this manuscript, the authors achieved a more complex cyclized skeleton through the enzyme-catalysed cyclization of linear terpene skeletons. The subsequent chemical derivatization of this skeleton produced a range of valuable compounds, thereby demonstrating the efficacy of the cyclizing enzyme. The enzyme-catalysed reaction offers significant advantages over chemical reactions, including higher selectivity and fewer by-products. The synthetic method developed in this study provides an innovative approach for future organic chemical synthesis and the creation of active compounds.

In addition, in response to the results of the previous round of peer review, the authors have conducted additional analyses and experiments that have satisfactorily addressed my primary comments and concerns. I recommend acceptance of this manuscript.

Answers to Reviewer 2:

At this point, however, this referee believes that the final area where additional improvement is still needed is with the spectra. As noted by two reviewers, several impurities existed within the NMR spectra of the original submission. That situation remains now, noting compounds 24, 14, S1-9, and 11 as just a few of many that have clear additional materials as a result of several non-enumerated peaks. Critical to this referee is that the authors comment on their yield calculations based on these impurities and/or delineate their presence directly in some way.

We agree with the reviewer that some of the compounds still contain minor impurities. We further improved the ¹H and ¹³C quality for compounds 3, 8, 11, 14, 18 and 24. However, for the side product S1-9 or intermediates P1-3, we couldn't get rid of the minor impurities without losing the material. We hope that the reviewer agrees with us that their structural elucidation by NMR is sufficient to draw our conclusions in the paper. We also changed the term isolated yield to crude yield for these three molecules (see Fig. S16 and pages 40-41). We also hope that the reviewer agrees with us that the incomplete full isolation of these products does not interfere with the main story of the paper.

Regarding the mix fraction of 11: This is not the isolated product, but the crude NMR of the geranyl geraniol biotransformation. It is just shown to elucidate the other products that occur during the biocatalytic cyclization.

Regarding determination of the NMR yield of 29 using mesitylene as an internal standard. To our understanding this is a routinely used method to determine the NMR yield. This is also mentioned in the paper.

We hope to satisfy the reviewer with these improvements.

Some points of confusion in the experimental remain, such as on page 44 the formation of the acetate makes little sense, especially as 1:1 impurity. The authors could merely have quenched their reaction differently, and or monitored by TLC, to ensure that an intramolecular cyclization proceeded smoothly, noting again that such products can arise directly via carbamate groups attached to the terminal alcohol of the starting materials.

We agree with the reviewer that our experiment did not achieve the best possible outcome. Indeed, the reaction has been monitored by TLC, however, quenching with ethyl acetate seemed to be the issue here. We changed the yield to NMR-yield in Figure 4.

We also agree with the reviewer that another starting material could have been used so that the reaction would proceed smoothly as demonstrated by Snyder and co-workers. However, the main focus of the paper was to demonstrate the simple applicability and tunability of biocatalytic cyclizations of naturally available linear starting material such as 6, 8, 10, 13, 17 and Z-17. We believe that our findings now enable the experimenter to choose between linear substrate derivatization before cyclization or product derivatization after cyclization. In this case we agree that prior derivatization would be presumably much more beneficial if the substrate fits into the active site of the enzyme. We hope to satisfy the reviewer with our comments.

Finally, it is also recommended the authors consider the precedents from Yamamoto and Corey for chiral Bronsted acid-promoted cyclizations as part of their opening analysis, not just the List paper, as many examples of this type exist and has been a problem that others were true pioneers in terms of initial examples. Better review citations might help here in the beginning, as again the authors ignore a lot of key scholarship.

We understand the reviewer's concerns but must kindly disagree with this point. We acknowledge the pioneering work of Yamamoto and Ishihara, and the work is already cited in the main text (reference 8). However, to our understanding, stereocontrolled cyclization of unbiased, abundant terpenes such as 6, 8, 10, 13, 17 and Z-17 is still a challenge using classical chemical methods (see Table S1). All the reviewer's provided references use strategic substrate modifications in the form of strong terminating nucleophiles, e.g. arenes, to successfully perform cationic cyclizations. Therefore, adding these examples would contradict the main intention of the paper (see also Fig. S1b; see ref. 5). By the way the cited review article (ref. 5) summarizes all the papers mentioned by reviewer 2. We hope that we have been able to satisfy the reviewer's concerns with our statement.

REVIEWERS' COMMENTS

Reviewer #2 (Remarks to the Author):

This revised manuscript is a further improvement on the previous submission, and in the opinion of this referee is now suitable for acceptance. No further adjustments are needed to this manuscript, and this referee believes that it has been substantively improved to tell its overall scientific study.